# Estimating Physical Activity Energy Expenditure from Video

Gayatri Kasturi[1*], Pragya Shrestha[1*], Scott J. Strath[2], and Rohit J. Kate[1†]

[1]Computer Science, University of Wisconsin-Milwaukee, Milwaukee, WI, USA
[2]Kinesiology, University of Wisconsin-Milwaukee, Milwaukee, WI, USA

*Abstract*—**It is well established that health and well-being greatly depends on a person's amount of daily physical activity. Hence accurate measurement of physical activity has been deemed critical in assessing an individual's current health state as well as in predicting future health issues. However, despite its importance, a method for accurately measuring physical activity has remained elusive due to various reasons ranging from technological to individual's adherence and compliance in the measurement task. In this work, a novel automatic method is presented to measure the amount of physical activity directly from video thus bypassing many of the common barriers. Physical activity is measured in terms of the energy expended by a person per minute relative to their resting state using the standard units of metabolic equivalents (METs). The method uses a three-dimensional convolutional neural network trained on videos that captured each subject doing activities of daily living while inside a whole-room calorimeter. The whole-room calorimeter provided gold standard energy expenditure values corresponding to each minute of the video which were used as targets for training the model. The experimental results using leave-one-subject-out cross-validation with seventeen subjects, each with twelve hours of video, showed that the method accurately estimated physical activity energy expenditure with overall root mean squared error of 0.71 METs per minute. This presents promising results to predict physical activity energy expenditure from video and warrants future studies that could be carried out in free-living naturalistic settings.**

*Keywords— physical activity assessment, energy expenditure, 3D convolutional neural networks, deep learning*

## I. INTRODUCTION

The amount of daily physical activity directly relates to a person's health, well-being and longevity [1,2]. Conversely, lack of physical activity has been found to lead to numerous adverse health outcomes and is also a major risk factor for several diseases [3,4]. Hence measuring the amount of physical activity is of paramount importance for assessing a person's health status as well as to make recommendations for preventing future illnesses [5]. This is especially important at places such as nursing homes and recovery and rehabilitation centers where a certain amount of physical activity may be prescribed to a patient and hence measuring it may be medically necessary for ascertaining compliance.

Despite the importance of measuring the amount of physical activity, currently there is no satisfactory method for measuring it. While self-report questionnaires provide ease of use, they are often marred with self-report sociability biases [6]. Body-worn accelerometers have recently emerged as tools for quantitatively assessing physical activity [7]. They measure the amount of acceleration over time which is then mapped to energy expenditure, typically using machine learning methods. However, heterogeneity in study design, data processing methods, and device wear location have led to large errors in estimating physical activity behavior metrics from these body-worn devices. As such, disparate interpretations of the amount of physical activity required from device-based estimations to promote health have recently been published, complicating definitive public health recommendations [8,9].

In this work, we present an alternate method to quantitatively estimate the amount of a person's physical activity. The method simply uses video observation. From the video, the method then estimates the energy expended by the person using deep learning, specifically a three-dimensional (3D) convolutional neural network (CNN) [10]. The network is trained using gold standard targets obtained for video recorded in a whole-room calorimeter [11]. The trained model can then be used to estimate energy expenditure only from a video. Unlike body-worn devices, such as accelerometers, the camera is detached from the person which makes it less invasive and more practical to assess free-living activity behaviors. This method is particularly suitable for monitoring a patient's physical activities inside a room/building at places such as nursing homes and rehabilitation centers as well as is useful for emerging fields such as smart fitness [12]. To the best of our knowledge, no previous work had used 3D CNN for predicting energy expenditure from videos with precise ground truth energy expenditure values obtained using a whole-room calorimeter. Our experimental results using data of seventeen subjects, each with twelve hours of video, show that the method works well.

## II. RELATED WORK

Although a significant amount of research has been done in recognizing the type of human activity from video [13-15] as surveyed in [16], as well as in estimating energy expenditure using wearable devices [31-33], relatively much less work has been done in estimating energy expenditure from video. In one such work, Peng et al. [17] collected several video clips from YouTube and other public sources and then annotated them with energy expenditure labels based on available physiological models and MET tables [18]. They compared deep learning methods on this dataset, but it should be noted that the energy expenditures in their dataset are only approximations which

---

[*] *Alumni at the time of submission.*
[†] *Corresponding author – Email: katerj@uwm.edu*

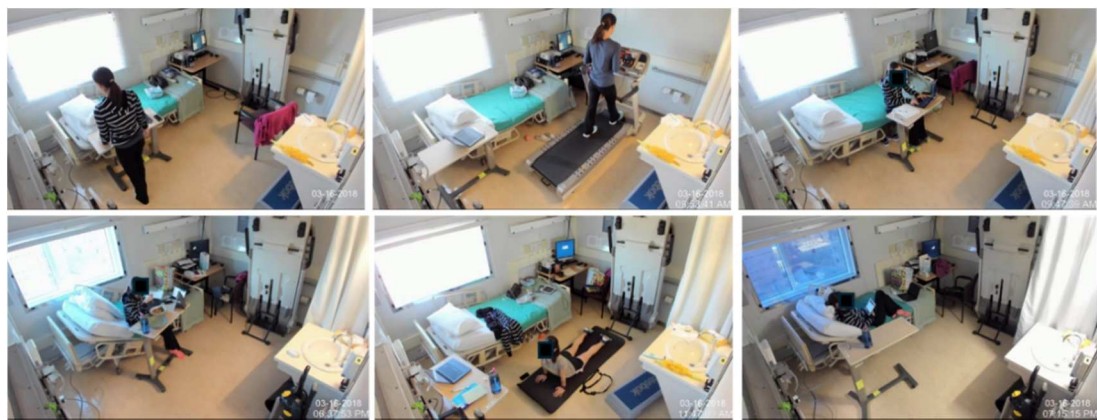

Fig. 1. Screenshots taken from a 12-hour video of one subject doing different physical activities. Face is shown covered with black rectangles for anonymity.

were retrospectively added, and unlike in our dataset, are not the actual values measured while performing the activities.

Nakamura et al. [19] do activity detection and energy expenditure estimation using multimodal signals from heart rate monitors, accelerometers and videos. Their videos were captured by camera worn by subjects while performing the activities. They also did not measure the actual energy expenditure ground truth values but retrospectively added them to their dataset based on the type of the activity and MET tables [18].

In Perrett et al. [20] (as well as in earlier studies by some of the same authors [21,22]) the authors used a wearable indirect calorimetry device to measure the ground truth energy expenditure for subjects performing various activities. They used a temporal convolutional neural network to predict energy expenditure from silhouettes of the subject (i.e. foreground segmentation) in the video. The physical activities were from a predefined set of activities such as stand, sit, vacuum, etc. Their data had 10 subjects and the total time duration was 4.5 hours. Our dataset differs from theirs in multiple ways. Instead of a wearable calorimetry device, we used a whole-room calorimeter which allows the user to be freer thus better simulating free-living conditions. The subjects in our data were not restricted to only a pre-defined set of activities but were also free to do anything they wanted. Our data is also much larger with 17 subjects and a total time duration of around 204 hours. Our machine learning approach is also very different from their approach.

## III. METHODS

### A. Dataset

Our dataset consisted of approximately 12-hour continuous video of each subject inside a whole-room calorimeter, for a total of 17 subjects. All subjects were healthy, and varied in age (22-74 years, with average age of $40.94 \pm 18.4$ years) and gender (7 male and 10 female). The subjects were provided a list of activities that they were asked to complete for a minimum of 5-minutes duration during their stay. This included computer work, reading, exercising, washing dishes, dusting, and sweeping/mopping. Outside of these activities the subjects were permitted to do anything they wanted. The room was self-contained for a subject to comfortably spend 12 hours. The

video camera was fixed at one corner of the room which captured full view of the room. Videos were recorded at 1280x720 pixel resolution and at 30 frames per second. Each 12-hour video of a subject was around 13 gigabytes in file size. Fig. 1 shows a few screenshots from a video of one subject. In this figure the face has been covered with black rectangles for anonymity.

Physical activity energy expenditure was measured using whole-room indirect calorimetry. Oxygen ($O_2$) and carbon dioxide ($CO_2$) concentrations were measured continuously using a fuel cell-based dual channel $O_2$ analyzer (FC-2 Oxzilla, Sable Systems, International, Las Vegas, NV) and two infrared $CO_2$ analyzers (CA-10 $CO_2$ analyzers, Sable Systems, International., Las Vegas, NV). $O_2$ consumption ($VO_2$) and $CO_2$ production ($VCO_2$) were calculated in 1-minute intervals using the flow rate and the differences in $CO_2$ and $O_2$ concentrations between entering and exiting air. The accuracy and precision of the system is confirmed monthly using propane combustion tests. The average $O_2$ and $CO_2$ recoveries are historically $\geqslant 98.0\%$.

The amount of energy a person expends to do a physical activity depends on the body weight. Hence to keep energy expenditure values independent of the weight of a person, the measured minute-by-minute energy expenditure values were converted into metabolic equivalent (MET) values (1 MET is equivalent to resting energy expenditure per unit body weight, or 3.5 ml/kg/min of oxygen consumption). MET is a standard unit of energy normalized for body weight. Given that same amount of energy per unit body weight is needed to move in the same motion, other subject parameters are not important in predicting energy expenditure in METs from motion (an unfit person may feel more tired than a fit person, but the energy spent by both for the same motion will be the same per unit body weight). MET is also a current standard anchor to measure energy expenditure in the exercise science and public health field, and is linked to exercise prescription, promotion, and health outcomes. Whole-room energy expenditure values were then used as targets to train our machine learning method to estimate minute-by-minute MET values from every minute of video observation. In our dataset, the average energy expenditure per minute over all the subjects was 1.7 METs with a standard deviation of 0.91 METs. Fig. 2 shows the distribution of energy expenditure in our data over all the subjects. It can be

seen than most of the energy expenditure was between 1-2 METs indicating that the subjects were not very active most of time which is typical of the contemporary sedentary lifestyle of most people and will be also typical for application domains such as nursing homes and rehabilitation centers.

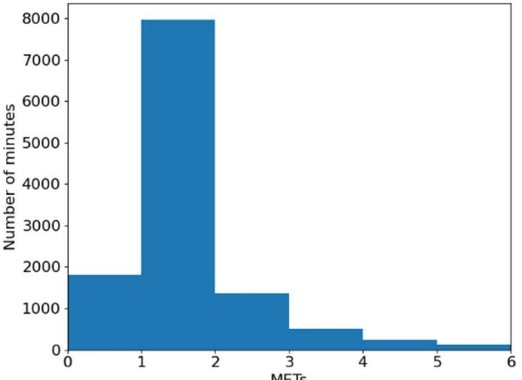

Fig. 2. Histogram showing energy expenditure distribution in the data.

### B. Machine Learning Approach

*Deep Learning Architecture:* We built a model to predict energy expenditure of a person from one minute of video. Hence the input to the model is one minute of video and the output is energy expenditure in METs. Convolutional neural networks (CNNs) are deep neural networks designed to work for grid like input data such as images, and have proven to be very effective for prediction tasks with images as input [23]. Typically, 2-dimensional (2D) CNNs are used for images as input. However, a video is a sequence of image frames hence a 3-dimesnional (3D) CNN is well-suited for a prediction task with videos as input in which time is treated as the third dimension. This enables the network to also learn temporal dependencies in the data. Therefore we built our model using a 3D CNN.

As mentioned earlier, the video data was captured at 1280x720 pixels with 30 frames per second. However, we found working with these dimensions computationally too demanding, especially with 12 hours of data per subject. This is also more granular than necessary for predicting energy per minute. Hence we reduced the frame size to 80x45. In our pilot experiments using a subset of data, we found that this frame size is sufficient for our task and the performance does not change much even

when using 320x180 frame size. Besides reducing the frame size, we also used a lower frame rate of one frame per second instead of original 30 frames per second. Given that a person does not move a lot within one second, this rate is sufficient to predict energy expenditure. Hence an input of one minute video goes in as 60 frames of size 80x45 to our 3D CNN model. For such input the model has to predict the corresponding energy expenditure value. The red, green and blue components of each pixel are treated as three channels of CNN, as is commonly done.

The 3D CNN architecture of our model is shown in Fig. 3. The 3D input of size 60x80x45 with three channels is followed by a 3D convolution layer with 32 filters of size 3x3x3 with "same" padding. This is followed by a 3D average pooling layer of size 2x2x2 which halves the lengths of every dimension. We intentionally chose average pooling instead of max pooling because in this task the network needs to capture all movements in the image frames in order to estimate the expended energy, but max pooling would tend to ignore some of the movements which are not maximum values in the pixel neighborhood. We repeat the combination of 3D convolution layer followed by 3D average pooling layer two more times.

At this point, the size of the data is 7x10x5. Next, a 3D convolution layer with 32 filters of size 1x10x5 and "valid" padding is added, that is, with the same filter size as the size of the image at this point. The reason for this is that because the camera is at one corner of the room, any movement near it will manifest as a large change between image frames; conversely, any movement far from it will manifest as small change between image frames. Hence this task does not conform to translation equivariance, a property of CNN layers, that treats every part of the image equally which is achieved through weight sharing. By giving a filter size as big as the frame size at this point, the CNN layer has to use different weights for different parts of the image so that it can learn the relation between movement in image frames and energy expenditure separately for different parts of the image. We do not do this in the earlier layers because it would then dramatically increase the number of weights to learn because of the much larger image size in earlier layers. Additionally, it is better to let the earlier layers abstract away more granular features from the image.

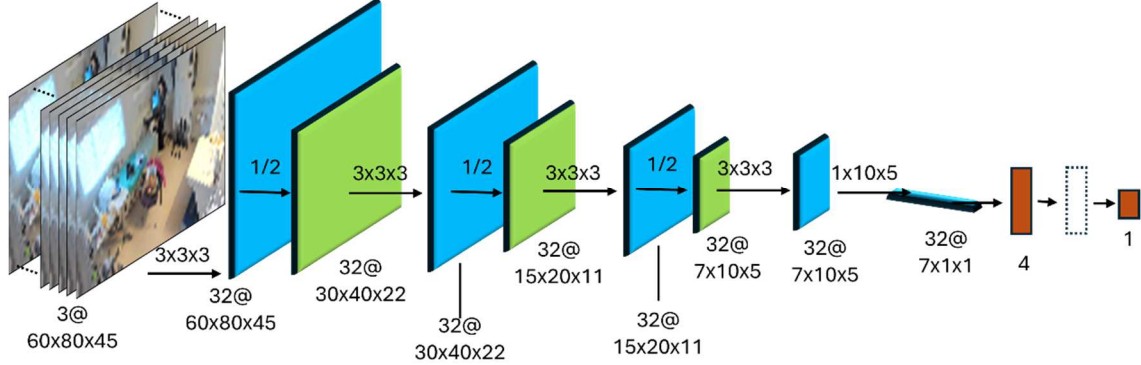

Fig. 3. The 3D CNN architecture used to estimate energy expenditure from one minute video. Blue rectangles are 3D convolution layers, green rectangles are 3D average pooling layers, brown rectangles are dense layers, and the white rectangle is a dropout

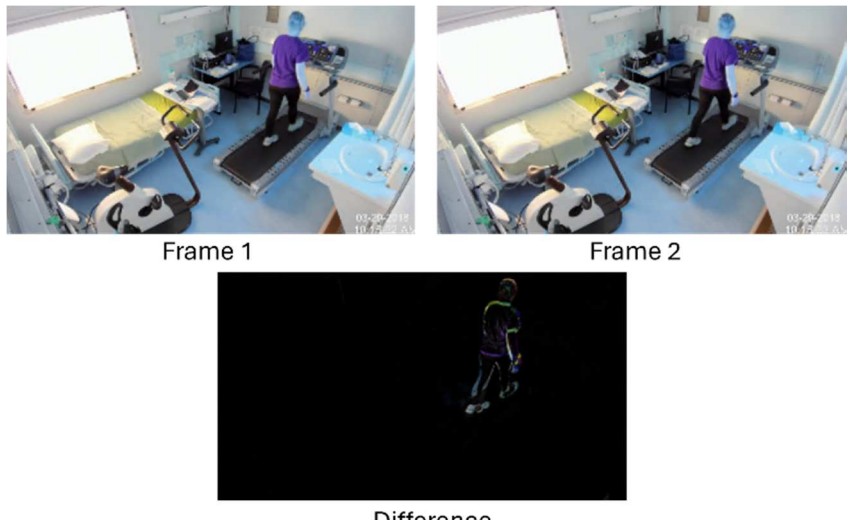

Fig. 4. Two consecutive frames and their difference frame. The non-black portion in the difference frame indicates change or movement between the original frames.

The next layer is a dense layer with four nodes followed by a dropout layer with 0.1 parameter value which is added to prevent overfitting. Finally, a dense layer of one node with linear activation is used that outputs the energy expenditure. Except for this last layer, leaky ReLU activation is used in all other layers.

Although deep learning methods, such as CNNs, can directly learn to predict from raw image frames, for our task it is clear that energy expenditure depends more on the change from one frame to another as the subject moves. Furthermore, more change between frames should correspond to more energy expenditure, while less or no change should correspond to less energy expenditure. It is possible that a 3D CNN can learn to pick this during the training process, however, it is also possible that it will waste its time during the training process focusing on various stationary objects in the room before it learns to isolate the person and then focus on the person's movement between frames. Hence besides using the raw frames, we employed two methods, described below, that directly provide changes between frames as input to the model.

*Method Using Frame Differences:* In this method, instead of giving the sequence of image frames (extracted from the video) as input, we give the difference between them, where the difference is simply pixel-wise subtraction for each of the red, green and blue components. Thus for 60 frames for one minute of video there will be 59 difference frames. If there has been no movement between the frames, the difference will be all zeros. Analogously, large movements will be captured by large values. Hence the difference between frames is likely to simplify the energy expenditure estimation task for the model. Fig. 4 shows an example of two frames and their computed difference frame. The difference frame clearly indicates with the non-black regions how the person moved between the two frames.

*Method Using Optical Flows:* While difference is a simple and straightforward way to compute the change between two

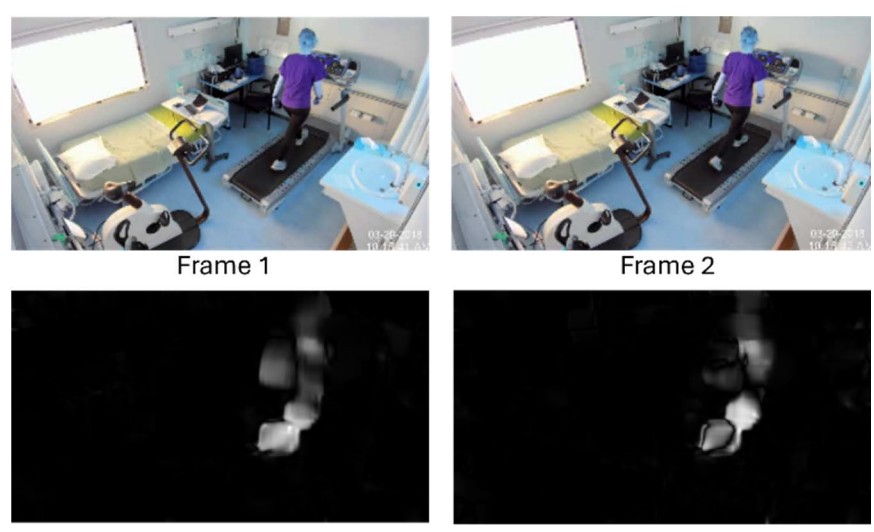

Fig. 5. Two consecutive frames and their dense optical flow magnitudes and angles.

frames which can be used as a proxy for the motion of objects, there exists a more principled method that captures the pattern of motion of objects between two image frames. This is called optical flow [24] which depicts the movement of points between two frames. Optical flow for all points (i.e. all pixels) between two frames is called dense optical flow which shows how every pixel moved from one frame to another. We used the well-known Farneback algorithm [25] to compute dense optical flow which is computed as the magnitude and angle (i.e. direction) of motion for each pixel. We used these values as the two channels for our 3D CNN. As with the method using frame differences, for 60 original frames for one minute of video there will be 59 optical flow frames. Fig. 5 shows an example of two frames, and their computed dense optical flow magnitudes and angles.

We show in our results that both the methods using frame differences and optical flows perform significantly better in predicting energy expenditure compared to directly using raw image frames.

### C. Experimental Methodology

*Cross-validation:* For experimental evaluation, leave-one-subject-out cross-validation was done using the data of 17 subjects described earlier. In this cross-validation, data of one subject was used for testing the model which was trained using the data of all the remaining subjects, and this entire process was repeated 17 times with a different test subject each time. The results were then averaged over these 17 folds and reported. Subject-wise cross-validation faithfully estimates how the system will perform on new subjects it will encounter when deployed.

*Evaluation Measures:* Estimating the numerical values of energy expenditure is a regression task hence the standard evaluation metric for regression, root mean squared error (RMSE), was used. This metric computes square root of the mean of the squares of prediction errors over all test examples, as depicted in equation (1), where n is the number of test examples. The lower the RMSE, the better the performance, with zero RMSE being the perfect performance.

$$RMSE = \sqrt{\frac{\sum_{i=1}^{n}(Target_i - Predicted_i)^2}{n}} \qquad (1)$$

An advantage of RMSE is that it is in the same units as the predicted quantity which makes it easy to interpret. In our results, RMSE is in METs and hence can be interpreted directly as the error in METs in estimating energy expenditure. However, one disadvantage of RMSE is that it is a domain-dependent relative measure and its exact numerical value (small or large) is not indicative of the performance in absolute terms. For example, when the target values are numerically large for a domain then a large RMSE may not be necessarily bad; on the other hand, when the target values are numerically small for a domain then a small RMSE may not be necessarily good. Hence besides RMSE, we used the evaluation metric of R-squared ($R^2$) which measures regression performance in absolute terms independent of the domain. It compares the performance of the model on the test set with the performance of a model that always predicts the average target value from the test set, as depicted in equation (2), where *n* is the number of test examples

and $\overline{Target}$ is the average of test target values. The higher the $R^2$, the better the performance, with $R^2$ value of one being the perfect performance. An $R^2$ value of zero indicates that the prediction model is not any better than a model that simply predicts the average target value.

$$R^2 = 1 - \frac{\sum_{i=1}^{n}(Target_i - Predicted_i)^2}{\sum_{i=1}^{n}(Target_i - \overline{Target})^2} \qquad (2)$$

*Implementation Details:* As described earlier, our dataset has around 12 hours of video for each subject which leads to around 720 (12x60) one-minute examples for each subject, hence around 11,520 (720x16) examples for training for each cross-validation fold. Given that each example itself consists of 60 image frames (59 in case of difference and optical flow based methods), we found training with the entire available training data computationally extremely demanding. But we found that it is not necessary to train with the entire data because the learning curve plateaus with 10% of the training data (a learning curve is later shown in the Results). Hence 10% of the training data was found sufficient for learning to estimate energy expenditure in our dataset. In order to sample 10% examples during training, our method used every 10th example (or every 10th minute) from the training data for each subject. Testing was, however, done on all the available test examples.

We used Python computer vision library, OpenCV [26], to process videos and image frames as well as to compute dense optical flows. To compute dense optical flows, the library's implementation of the Farneback's algorithm was used whose parameter values were set based on a tutorial [27]. We used deep learning Python library Keras [28] to implement 3D CNNs. Mean squared error was used as the loss function and Adam [29] as the optimizer. For training the models, 15 epochs were found to be sufficient after which the training loss barely decreased. All other parameter values were kept at their default values.

## IV. RESULTS AND DISCUSSION

Table I shows the results of estimating energy expenditure obtained using leave-one-subject-out cross-validation as described in the previous section. The table compares the results of directly using the original image frames, using the frame differences, and using the optical flows. For each case, a different model was trained but the same 3D CNN architecture was used. The table shows RMSE (in METs) and $R^2$ means and standard deviations computed over the 17 subjects used in leave-one-subject-out cross-validation. The best result is shown in bold and italics. The result which was not found to be statistically significantly different from the best result (as determined by two-tailed paired t-test using p-value < 0.01) is shown in bold.

TABLE I
RESULTS OF ESTIMATNG ENERGY EXPENDITURE

| Method | RMSE Mean (Std. dev.) | $R^2$ Mean (Std. dev.) |
|---|---|---|
| **Original Frames** | 0.88 (0.42) | -0.12 (0.21) |
| **Frame Differences** | **0.74 (0.37)** | **0.20 (0.28)** |
| **Optical Flows** | ***0.71 (0.34)*** | ***0.23 (0.35)*** |

It can be observed from Table I that optical flow based method obtained the best result with mean RMSE of 0.71 METs. This was closely followed by the difference based method with mean RMSE of 0.74 METs which was not found to be statistically significantly different from the optical flow based method's result. Given that 1 MET corresponds to resting energy expenditure, these RMSE values are good. Directly using the original image frames obtained the worst result with mean RMSE of 0.88 which was statistically significantly different from the result of each of the other two methods. Its mean $R^2$ is negative indicating that on average it did not perform any better than simply predicting the average test target value (but note that this average value is not known for any test subject during testing, hence simply predicting the average test target value cannot be considered as a legitimate baseline prediction method). This shows that directly using image frames does not always lead to meaningful learning of the relation between video and energy expenditure. On the other hand, both the difference and the optical flow based methods are able to capture the relation as indicated by their lower RMSE as well as positive mean $R^2$ values.

We note that the RMSE obtained in this study for estimating energy expenditure using videos falls in the range obtained by other studies using wearable devices (RMSE: 0.5-1.6 METs) [31-33], although these are not directly comparable to our results because the datasets as well as the data collection settings were different.

For a closer look at the results, Fig. 6 shows subject-wise RMSE (in blue) obtained using the optical flow based method which obtained the best results. For comparison, the figure also shows the mean energy expenditure (in orange) of each subject along with its standard deviation (black line). It can be seen from the figure that the subjects varied greatly in how active they were in the 12-hour period. For example, subjects with IDs 4, 16 and 17 were overall more active than others as well as had more variation in their activity levels leading to higher mean energy expenditure and standard deviation. In contrast, subjects with IDs 10 and 12 were relatively less active. It can be observed that subject-wise RMSE was generally proportional to the mean energy expenditure level of the subject and its standard deviation. This is not surprising because lower energy expenditure level indicates that the subject was sedentary for a majority of the time which makes it easier to predict their energy expenditure. On the other hand, higher energy expenditure level indicates that the subject was moving a lot, making it harder to accurately predict the energy expenditure. Furthermore, given that these subjects were more active than the rest, it makes their data different from the majority of the subjects on which the model was trained, thus leading to an inferior performance on these subjects. This indicates that in future, training may benefit if subjects with more varied levels of activity are included, especially some with higher activity levels.

As an example of actual predictions over the entire 12-hour period, Fig. 7 shows the correct energy expenditure values for one of the subjects (Subject 8) and the values predicted by the model which was trained on the data of all the remaining subjects using the optical flow based method. It can be observed that for the most part, the model was able to make predictions very close to the correct values, however, it made prediction errors at some extreme values.

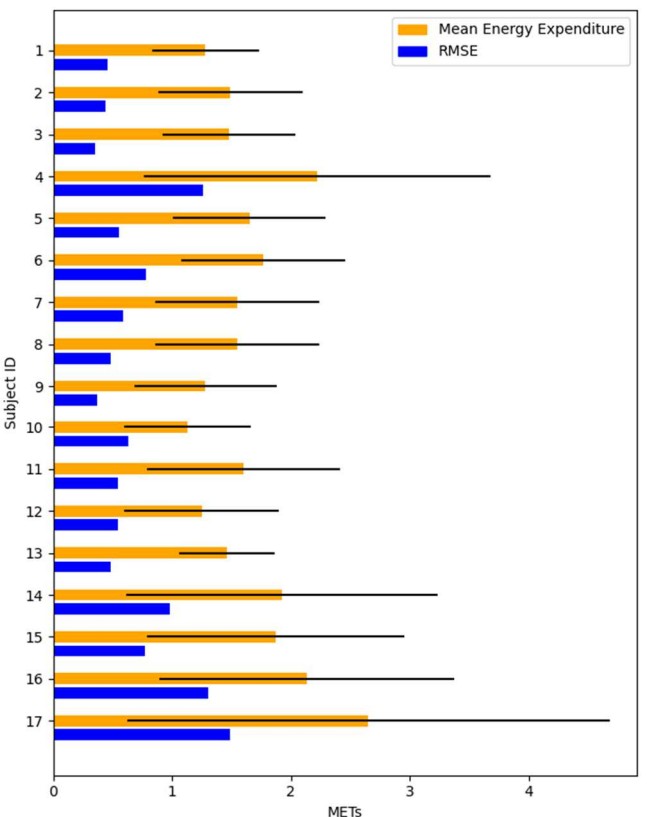

Fig. 6. Subject-wise mean energy expenditure with standard deviation and RMSE obtained using the optical flow based method.

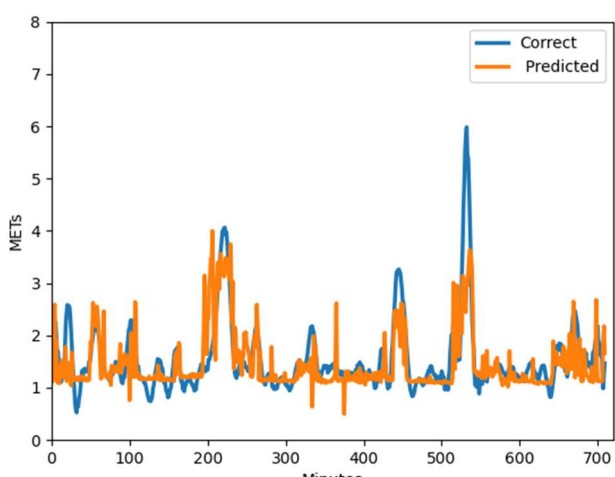

Fig. 7. Correct and predicted energy expenditure values for a subject over 12-hour period.

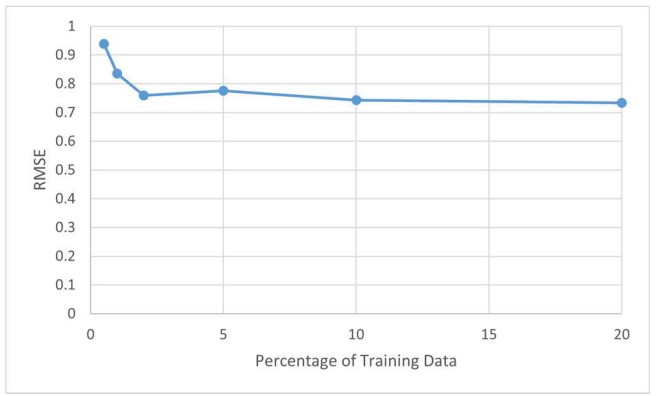

Fig. 8. Learning curve obtained using increasing amount of training data for the frame difference based method.

Fig. 8 shows a learning curve depicting the performance with increasing amount of training data obtained using the frame difference based method (optical flow based method gives a similar trend). To obtain this curve, leave-one-subject-out cross-validation was done but using only a fixed percentage of training data every time. Testing was done on the entire available data for the test subject in each fold. As was pointed out in the previous section, the performance nearly plateaus with 10% of the training data (which is already more than 1,100 examples) and using more data is not helpful but only computationally more demanding. This also indicates that increasing the size of the training data by making the subjects stay in the room for longer durations will not benefit the performance of the method.

On a good desktop computer (Intel Core i7-10700T 2GHz processor with 16 GB RAM), for each fold of cross-validation, it took 7.9 minutes on average to train (with 10% of the training data) and 7 seconds on average to test. Given that it takes only 7 seconds to estimate energy for 12 hours of data, the trained system has the potential to be deployed in real time to do the energy estimation task.

## V. LIMITATIONS AND FUTURE WORK

One limitation of this work is that the video data always had one person in the room whose energy expenditure was to be estimated. If the room had other people then the model will not work as desired. Additionally, if there were moving objects in the room (for e.g., a ceiling fan) then the model will likely be affected. In future, a method could be used to first isolate and track the person of interest [30] and then apply the model. Another limitation is that the same room was used for all the subjects. This was because it is very expensive to create a whole-room calorimeter. The performance of the model could possibly degrade if a test subject is in a different room environment, although a model that uses optical flow or difference between frames is less likely to be affected by the room because it would only focus on the changes between the frames, as shown in Fig. 4 and Fig. 5. However, in future, the method should be also tested in varied room environments as well as in more naturalistic settings. Finally, including more subjects with varied activity levels for training will likely lead to a model that generalizes better across subjects.

## VI. CONCLUSION

Estimation of energy expenditure is important for a person's health assessment, especially at places such as nursing homes and rehabilitation facilities. In this work, we presented an automatic method for energy expenditure estimation using only a person's video. The method used 3D CNNs to predict energy expended by a person in every minute of the video. The method was trained and tested using data with precise ground truth energy expenditure values obtained using a whole-room calorimeter. Experiments were conducted using a large video data of 17 subjects, each of 12-hour duration. We also introduced methods based on the difference between image frames and optical flows which were shown to perform better than directly using the original frames from the videos. Results showed that the method is good at estimating physical activity energy expenditure from video. Future study designs could include more varied and naturalistic settings.

### ACKNOWLEDGMENT

Research reported in this publication was supported in part by the National Cancer Institute of the National Institutes of Health award number R01CA215318-Strath.

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
