# OpenReview forum: "Estimating Physical Activity Energy Expenditure from Video"
_IEEE.org/EMBS/BHI/2024/Conference — IEEE BHI'24_

### Official Review · Reviewer_euLR · 2024-08-10
**Promising approach for estimating energy expenditure from video through deep learning and motion estimation approaches**

**Overall Rating:** 8
**Confidence:** 5

**Other Quality Metrics:**

(a) Clarity of writing                : GREAT, paper is well-organized and written in a fluent language
(b) Clinical Significance          : GOOD, proposed energy expenditure estimation scheme is promising, simple and easily applicable, however additional training using more subjects, different activity scenarios and examining human body characteristics are required for extracting robust and precise results
(c) Methodological Novelty     : FAIR, innovative characteristics of presented study are mostly related to the application field and training with specially created dataset, algorithmic scheme seems to be directly adopted from already developed resources
(d) Experiments and Results  : GOOD, since incorporating motion estimation into image analysis seems to improve performance of initially developed framework, yet additional data are required for providing robust results and outcomes

**Questions For The Authors:**

In order to further improve the quality of the manuscript, the following amendments could be addressed:
* presentation of related work is compact, revealing limitations and challenges with respect to the examined scientific field, yet some references could be updated
* what is the novel contribution of the authors regarding the algorithmic setup and the deployment of the 3D CNN model for estimating energy expenditure from video?
* how are R,G and B color bands fused into one image sample for model training/testing. Any pre-processing step applied towards image enhancement and noise removal?
* how is ground truth incorporated into the training procedure? Is it a one MET value for the entire image sequence, or are instant values utlized for each frame (or groups of frames)? How are parameters such as age, sex, height, weight addressed since each human subject has a unique "metabolism and energy expenditure profile"?
* any estimation on time-efficiency of proposed framework? Could it be adopted in real (or near real) time scenarios?

**Strengths:**

Advantages of presented study can be summarized as follows:
* training of deep learning model using real data, simulating real physical activity environment
* promising preliminary results
* effective and easy to apply methodology based on image analysis and motion estimation, not on wearable sensors and complicated equipment frameworks
* potential for improved performance through training including different activity scenarios

**Summary Of The Paper:**

Thank the authors for the submission of the manuscript. The paper introduces a 3D Deep Learning based approach for quantitatively estimating physical activity energy expenditure from image sequence, which seems promising and can be improved using image features after isolating motion between successive frames. Based on an already established CNN architecture, the authors trained the model with real data created in properly formulated laboratory conditions.

**Weaknesses:**

Weaknesses of presented study can be summarized as follows:
* parameters such as sex, height, weight, age play an important role in estimating energy expenditure, which are not considered in the training of DL model
* validation is performed on a small dataset and under limited activity scenarios
* novelty of proposed methodology is not clearly indicated, since a 3D CNN architecture is directly applied, while implementation is based on existing Python packages adopting default algorithmic parameter values

---

> ### Author Rebuttal · Authors · 2024-09-02
>
> We thank the reviewer for the detailed feedback. Following are our responses.
>
> >> presentation of related work is compact, revealing limitations and challenges with respect to the examined scientific field, yet some references could be updated
>
> - Response: We are not sure which other references the reviewer is asking to include given that very little work has been done in prediction energy from vidoes, but we have add a few more references in the related work section of the revised version of the paper.
>
> >> what is the novel contribution of the authors regarding the algorithmic setup and the deployment of the 3D CNN model for estimating energy expenditure from video?
>
> - Response: As another reviewer mentioned, the novelty of this work is in integrating the modelling techniques for this particular application. To the best of our knowledge, no previous work had used 3D CNN for predicting energy expenditure from videos with precise ground truth energy expenditure values obtained using a whole-room calorimeter. We also incorporated optical flows to do it. We have made this explicit in the revised version of the paper (tracked changes in the Introduction).
>
> Additionally, in the description of the architecture we mentioned two specific things we did in order to adapt CNN architecture for our particular application. First, we intentionally used average pooling to enable 3D CNN to capture all movements in the frames instead of max pooling which would tend to ignore some of the movements which were not maximum values in the pixel neighborhood. Second, we used filters of the same size as the size of the input in the last CNN layer which is an unconventional thing to do. As explained there, we did so because this task does not conform to translation equivariance which CNN layers conventionally achieve when smaller filter sizes are used.
>
> >> how are R,G and B color bands fused into one image sample for model training/testing. Any pre-processing step applied towards image enhancement and noise removal?
>
> - Response: We used the standard technique used in CNNs to incorporate R, G and B components by using them as three channels of the input. Hence in the first layer, separate weights are learned for each color component and then the resulting outputs are added which go as input to the next layer. No preprocessing was done for the images extracted from the videos because they were found to be already good.
>
> >> how is ground truth incorporated into the training procedure? Is it a one MET value for the entire image sequence, or are instant values utlized for each frame (or groups of frames)?
>
> - Response: There was one MET value associated with every one minute of video which was a sequence of 60 frames. We have clarified this in the revised version of the paper (tracked changes on Page 3 second column and the preceding sentence).
>
> >> How are parameters such as age, sex, height, weight addressed since each human subject has a unique "metabolism and energy expenditure profile"?
>
> - Response: All energy values used in this study were in the units of metabolic equivalent (MET) which is computed per unit body weight of the subject. Hence MET already accounts for the subject’s body weight. And from physics point of view, same amount of energy per unit body weight is needed to move in the same motion, hence other subject factors such as sex, height, age, etc. are not important in predicting energy expenditure in METs from motion (an unfit person may feel more tired than a fit person, but the energy spent by both for the same motion will be the same per unit body weight). MET is also a current standard anchor to measure energy expenditure in the exercise science and public health field, and is linked to exercise prescription, promotion, and health outcomes. We have included this clarification in the revised version of the paper (tracked changes on Page 2).
>
> >> any estimation on time-efficiency of proposed framework? Could it be adopted in real (or near real) time scenarios?
>
> - Response: We thank the reviewer for bringing this up. On an Intel Core i7-10700T 2GHz processor with 16 GB RAM, it takes 7.9 minutes on average to train and 7 seconds on average to test for each fold. Given that it takes only 7 seconds to estimate energy for 12 hours of data, the trained system can be certainly deployed in real time. We have now mentioned this in the revised version of the paper (tracked changes on Page 7).

---

### Official Review · Reviewer_ZacL · 2024-08-11
**Estimating Energy Expenditure from Video**

**Overall Rating:** 5
**Confidence:** 5

**Other Quality Metrics:**

(a) Clarity of writing - good;
(b) Clinical Significance - fair;
(c) Methodological Novelty - fair;
(d) Experiments and Results - fair

**Questions For The Authors:**

Please compare the proposed approach with existing methods for energy expenditure estimation using wearable devices, such as
M. Sevil et al., "Determining Physical Activity Characteristics From Wristband Data for Use in Automated Insulin Delivery Systems," in IEEE Sensors Journal, vol. 20, no. 21, pp. 12859-12870, 1 Nov.1, 2020, doi: 10.1109/JSEN.2020.3000772.

How does the proposed approach incorporate temporal dependencies in the video frames?

Why not also assess physical activity modality in the videos?

**Strengths:**

Use of gold-standard whole-room calorimeter

Use of convolutional neural networks for video analysis

**Summary Of The Paper:**

An automatic method for energy expenditure estimation using only a person’ video is developed that uses convolutional neural networks to predict energy expended by a person in every minute of the video with precise ground truth energy expenditure values obtained using a gold-standard whole-room calorimeter.

**Weaknesses:**

No comparison to energy expenditure estimation using wearable devices

---

> ### Author Rebuttal · Authors · 2024-09-02
>
> We thank the reviewer for the detailed feedback. Following are our responses.
>
> >> Please compare the proposed approach with existing methods for energy expenditure estimation using wearable devices, such as M. Sevil et al., "Determining Physical Activity Characteristics From Wristband Data for Use in Automated Insulin Delivery Systems," in IEEE Sensors Journal, vol. 20, no. 21, pp. 12859-12870, 1 Nov.1, 2020, doi: 10.1109/JSEN.2020.3000772.
>
> -	Response: The addition of wearable devices would have been advantageous, and in future studies we plan on incorporating such devices. But for this current study, this data is not available to make a comparison. However, we point out that the RMSE obtained in this study using videos (around 0.7 METs) is in the same range as obtained by other studies using wearable devices, although these are not directly comparable because the datasets as well as the data collection settings are different. We have added this to the revised version of the paper (tracked changes on Page 6).
>
> >> How does the proposed approach incorporate temporal dependencies in the video frames?
>
> -	Response: The 3D CNN is given time as the third dimension, i.e. it takes time sequence of 2D frames as input. Hence the 3D CNN is able to learn temporal dependencies through its learnable weights. We have mentioned this in the revised version of the paper (tracked change on Page 3 first column).
>
>
> >> Why not also assess physical activity modality in the videos?
>
> -	Response: The focus of this paper was on estimating energy expenditure which is important for assessing a person’s health status. Hence the presented method was developed to directly estimate it from videos. Assessing the type of activity is a different task for which a significant amount of research has been already done as we mentioned in the Related Work section with citations [13-16].

---

### Official Review · Reviewer_N4pW · 2024-08-14
**Automatic measurement of physical activity vy interpreting video of subjects in a  whole-room calorimeter.**

**Overall Rating:** 6
**Confidence:** 5

**Other Quality Metrics:**

(a) Clarity of writing:  Well-written manuscript

 (b) Clinical Significance:  Valuable approach with clinical relevance for nursing homes, assisted living facilities and rehabilitation centers

 (c) Methodological Novelty:  The data collection method and the modeling technique are known and considered as reliable approaches.  The novelty is in integrating them for for this particular application.

 (d) Experiments and Results: Well designed data collection approach and experiments.   Data collection focusing on activities of daily living may be relevant for a segment of people in for nursing homes, assisted living facilities and rehabilitation centers.  Extending data collection to medium-intensity physical activity would have been useful for more active people who may reside in such housing or people undergoing rehabilitation with intensity increase over time.

**Questions For The Authors:**

No questions.  Suggestions for enhancing the contributions of the paper are given in the Weaknesses" section

The discussion in the manuscript on subject-wise cross-validation vs. example-wise cross-validation is confusing.  Is it necessary since subject-wise cross-validation is the approach that makes sense for this application.

**Strengths:**

The whole-room calorimeter provides gold-standard energy expenditure data.

The model developed with a 3-dimensional convolutional neural network provides reliable estimates

Comparison of energy expenditure estimates by using the original video frames, frame differences, and optical flows provides information on the advantages of using optical flows

The proposed method is suitable for monitoring the physical activities of a person inside a room such as nursing homes and rehabilitation centers (as stated by the authors)

**Summary Of The Paper:**

A new method to automatically measure the amount of physical activity directly from video is reported. Physical activity is measured in terms of the energy expended by a person per minute relative to their resting state using metabolic equivalent task (METs). The method uses a 3-dimensional convolutional neural network (3D CNN)  trained on videos of each subject doing activities of daily living while inside a whole-room calorimeter. The calorimeter provided gold-standard energy expenditure values corresponding to each minute of the video. The calorimeter data were used as targets for training the 3D CNN model. The models developed with experimental results by using leave-one-subject-out cross-validation with 17 subjects, each with 12 hours of video, showed that the method accurately estimated physical activity energy expenditure for people preforming activities of daily living.

**Weaknesses:**

Collecting video data in free living in open spaces would be challenging, but targeting nursing homes and rehabilitation centers is appropriate and very useful.

Cofactors such as the physical conditioning of subjects may affect the results

Assessing the potential to build individual models for each subject should be useful.  Mode data may be needed for each individual either by using sliding windows or extended data collection periods.  Personalized models would address the concerns stated on page 6 and information provided on figure 6.

A figure comparing the time series data from the  whole-room calorimeter with the estimates from the model over the whole data collection period would be informative.

---

> ### Author Rebuttal · Authors · 2024-09-02
>
> We thank the reviewer for the detailed feedback.
>
> >> Collecting video data in free living in open spaces would be challenging, but targeting nursing homes and rehabilitation centers is appropriate and very useful.
>
> -	Response: Thanks, we agree.
>
>
> >> Cofactors such as the physical conditioning of subjects may affect the results
>
> -	Response: All energy values used in this study were in the units of metabolic equivalent (MET) which is computed per unit body weight of the subject. Hence MET already accounts for the subject’s body weight. And from physics point of view, same amount of energy per unit body weight is needed to move in the same motion, hence other subject factors such as physical conditioning are not important in predicting energy expenditure in METs from motion (an unfit person may feel more tired than a fit person, but the energy spent by both for the same motion will be the same per unit body weight). MET is also a current standard anchor to measure energy expenditure in the exercise science and public health field, and is linked to exercise prescription, promotion, and health outcomes. We have included this clarification in the revised version of the paper (tracked changes on Page 2).
>
>
> >> Assessing the potential to build individual models for each subject should be useful. Mode data may be needed for each individual either by using sliding windows or extended data collection periods. Personalized models would address the concerns stated on page 6 and information provided on figure 6.
>
> -	Response: Thanks for noting this. It is certainly possible to build a personalized model for an individual subject by training it only on that subject’s data (and evaluating it by cross-validation within that subject’s data). In fact, such a model gives very good performance. But our system is meant to be deployed where it will need to predict energy expenditures for new subjects who would not have spent time earlier in the whole-room calorimeter for gathering their training data. Hence in this study we trained models on the data of multiple subjects so that the models could generalize well to new subjects when deployed.
>
>
> >> A figure comparing the time series data from the whole-room calorimeter with the estimates from the model over the whole data collection period would be informative.
>
> -	Response: Thanks for the suggestion. We have included such a figure in the revised version of the paper (Figure 7).
>
> >> The discussion in the manuscript on subject-wise cross-validation vs. example-wise cross-validation is confusing. Is it necessary since subject-wise cross-validation is the approach that makes sense for this application.
>
> -	Response: In the revised version we have removed the comparison (tracked changes on Page 5).

---

### Decision · Program_Chairs · 2024-09-23

Accept